# Prediction of viral symptoms using wearable technology and artificial intelligence: A pilot study in healthcare workers

Pierre-François D'Haese[1,2,3‡]*, Victor Finomore[1,2,3‡], Dmitry Lesnik[4],
Laura Kornhauser[4], Tobias Schaefer[4], Peter E. Konrad[1,2,3], Sally Hodder[1,2,3],
Clay Marsh[1,2,3], Ali R. Rezai[1,2,3]

1 Rockefeller Neuroscience Institute, West Virginia University, Morgantown, West Virginia, United States of America, 2 West Virginia Clinical and Translational Science Institute, West Virginia University, Morgantown, West Virginia, United States of America, 3 Health Sciences Center, West Virginia University, Morgantown, West Virginia, United States of America, 4 Stratyfy, Inc, New York, New York, United States of America

‡ PFD and VF are Co-first authors.
* pd0033@hsc.wvu.edu

**Data Availability Statement:** Data cannot be shared publicly because of parts are being owned by a third party. Data are available from the RNI

## Abstract

Conventional testing and diagnostic methods for infections like SARS-CoV-2 have limitations for population health management and public policy. We hypothesize that daily changes in autonomic activity, measured through off-the-shelf technologies together with app-based cognitive assessments, may be used to forecast the onset of symptoms consistent with a viral illness. We describe our strategy using an AI model that can predict, with 82% accuracy (negative predictive value 97%, specificity 83%, sensitivity 79%, precision 34%), the likelihood of developing symptoms consistent with a viral infection three days before symptom onset. The model correctly predicts, almost all of the time (97%), individuals who will not develop viral-like illness symptoms in the next three days. Conversely, the model correctly predicts as positive 34% of the time, individuals who will develop viral-like illness symptoms in the next three days. This model uses a conservative framework, warning potentially pre-symptomatic individuals to socially isolate while minimizing warnings to individuals with a low likelihood of developing viral-like symptoms in the next three days. To our knowledge, this is the first study using wearables and apps with machine learning to predict the occurrence of viral illness-like symptoms. The demonstrated approach to forecasting the onset of viral illness-like symptoms offers a novel, digital decision-making tool for public health safety by potentially limiting viral transmission.

## Introduction

Virus transmission from asymptomatic or pre-symptomatic individuals is a key factor contributing to the SARS-CoV-2 pandemic spread. High levels of SARS-CoV-2 virus have been observed 48–72 hours before symptom onset. As high viral loads of SARS-CoV-2 may occur before the onset of symptoms, strategies to control community COVID-19 spread that rely

Institutional Data Access / Ethics Committee (contact via Dr Padmashree Tirumalai, padma. tirumalai@hsc.wvu.edu) for researchers who meet the criteria for access to confidential data.

**Funding:** Stratyfy provided financial support for this study in the form of salaries for DL, LK, and TS. The specific roles of these authors are articulated in the 'author contributions' section. The funders had no role in study design, data collection and analysis, decision to publish, or preparation of the manuscript. No additional external funding was received for this study.

**Competing interests:** The authors have read the journal's policy, and the authors of the study have the following competing interests to declare: DL, LK, and TS are paid employees of Stratyfy. This does not alter our adherence to PLOS ONE policies on sharing data and materials. There are no patents, products in development or marketed products associated with this research to declare.

only on symptom-based detection are often unsuccessful. The development of novel approaches to detect viral infection symptoms during this pre-symptomatic phase are critical to reducing viral transmission and spread by facilitating appropriate early quarantine before symptoms occur.

Once infected, the incubation period commonly ranges from 2–14 days (mean of 5.2 days), and infectious transmission starts around 2.5 days and peaks at 0.7 days before the onset of symptoms [1–4]. Of note, the loss of sense of smell and taste are more specific symptoms for COVID-19 [3]. Even when symptomatic COVID-19 occurs, the symptoms and signs of COVID-19 overlap with other viral illnesses such as influenza.

Today, 1 in 5 Americans use fitness tracking devices [5]. While these technologies can inform population-level data sharing to detect disease state [6–9], to our knowledge, they have not been used to forecast communicable infectious disease at the individual level. Outputs from wearable technology including heart rate (HR), heart rate variability (HRV), respiration rate (RR), temperature, blood oxygenation, sleep, and other physiological assessments are increasingly being explored in studies of health and disease [10–12]. Moreover, a variety of subject-reported symptoms captured on mobile apps transforms both surveillance and contact tracing management strategies for COVID-19 [13–15].

Machine-learning algorithms are becoming more popular and useful when collecting large amounts of disparate data to provide insight into otherwise complex relationships not easily determined with routine statistical methods. Using a machine learning model informed by self-reported symptoms, we demonstrate that the combination of physiological outputs from wearable technology and brief cognitive assessments can predict symptoms and signs of a viral infection three days before the onset of those symptoms. This forecasting model could be used to enhance conventional infection-control strategies for COVID-19 and other viral infections.

## Methods

### Study design

The Rockefeller Neuroscience Institute (RNI) team initiated a study approved by the institutional review board (IRB) at the West Virginia University Medical Center (#2003937069), Vanderbilt University Medical Center (#200685), and Thomas Jefferson University (#2004957109A001) to combine physiological and cognitive biometrics and self-reported symptoms information from individuals at risk for exposure to COVID-19 and potential contracture of a viral illness. We recruited study participants from each tertiary medical center by approaching front-line health care workers receiving regional referrals for COVID-19 patients. We asked each participant to 1) wear a smart ring device [16] with sensors that collect physiological measures such as body temperature, sleep, activity, heart rate, respiratory rate, heart rate variability; 2) use a custom mobile health app [17] to complete a brief symptoms diary [3], social exposure to potentially infected contacts, and measures of physical, emotional, and cognitive workload; (see S1 Table and S1 File) as well as the psychomotor vigilance cognitive task (PVT) [18] to measure attention and fatigue twice a day. All data are collected, structured, and organized into the RNI Cloud data lake for analysis. The RNI Cloud is a HIPAA compliant data platform hosted in Amazon Web Services (AWS) that supports all the security and legal requirements to protect the data's privacy and integrity from the participants in the context of multi-center clinical studies [19].

We utilized a machine learning approach that combines features through probabilistic rules and provides a prediction. The training process consists of two steps. It combines subject reported symptoms (labeling model) to inform a predictive framework (forecast model) that uses physiological and cognitive signals to forecast suspicion of a viral illness (Fig 1). The

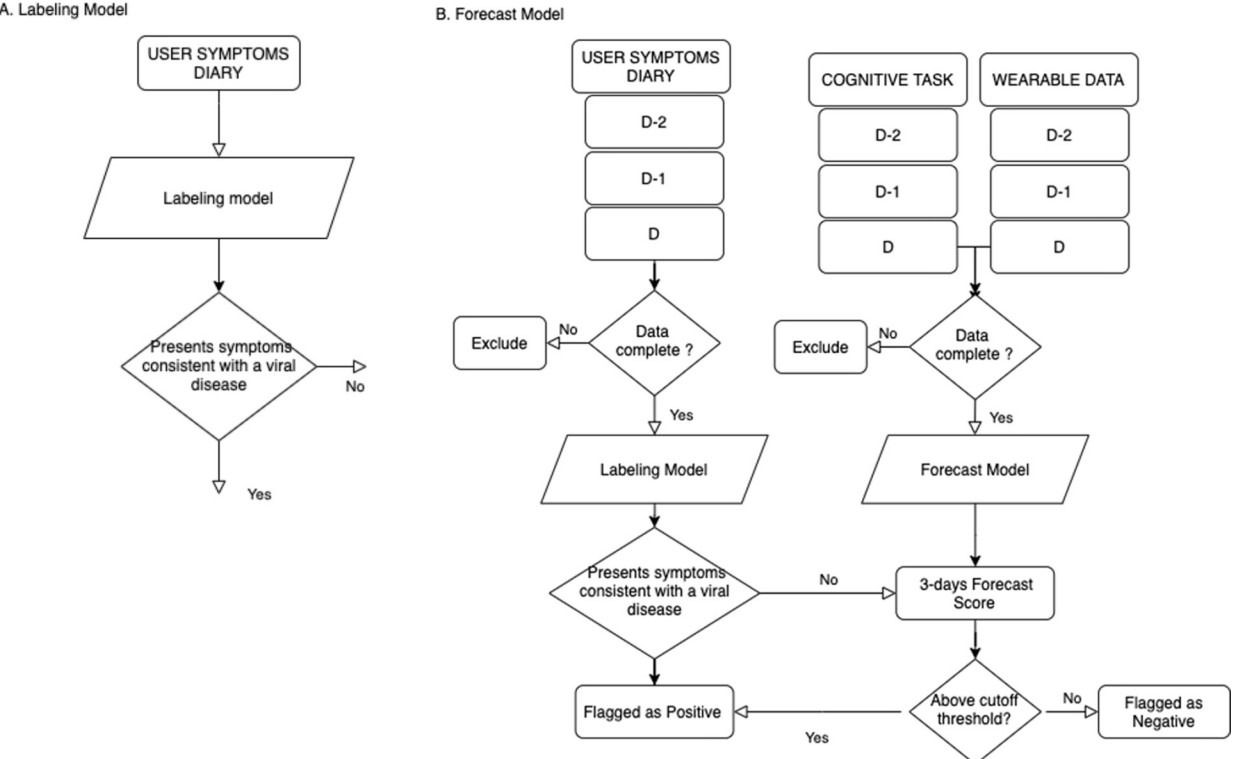

**Fig 1.** Data Flow a) the labeling model, b) the forecast model. Each model takes as input three days of data (d, d-1, d-2).

dataset consisting of PVT and wearable data is split, 75% for training, and 25% reserved for testing the model [20] (see S4 File). The labeling and forecasting models are created from a set of rules combining one or more features. All rules are given a weight and combined to provide a final decision [21–23] (see S2 Table).

## Labeling model

We use a rule-based approach to create an AI model that labels an individual's self-reported symptoms as suspicious (or not) for presenting symptoms consistent with a viral illness. The labeling model is created based on the expert knowledge manually translated into decision rules (see below). The purpose of this model is to define if a person is being suspicious of an infectious disease (below we just say *suspicious*) based on its self-reported symptoms. Rules are based on those symptoms commonly present in a diagnosed viral-like condition and those more specific for SARS-Cov-2 (e.g., loss of taste and smell) [2, 24, 25]. Resulting rules (Table 1) assign, for instance, higher confidence on suspicion of a viral-illness for self-reported fever with the persistence of symptoms for more than two consecutive days. In comparison, lower confidence is assigned for stuffy nose and swollen eyes without fever (see S5 File). The rules and weights in this model were establish from clinical subject matter experts. In particular, the weights associated with the rules were chosen to minimize the labeling error assessed by medical experts. We also fine-tuned some rule weights by fitting the model to a small synthetic data set, which contained typical symptom combinations. The actual calculations of the labeling model's output score is based on the machinery of the probabilistic logic, described in more detail in the next paragraph.

**Table 1. Labeling model rules.**

| Conditions that INCREASE suspicion of viral-like symptoms | THEN | Confidence | Comments |
|---|---|---|---|
| Are_you_Positive_for_COVID_19 | suspicious of viral-like symptoms | high | |
| Sense_of_Smell_Change | suspicious of viral-like symptoms | high | |
| Fever | suspicious of viral-like symptoms | high | |
| Cough | suspicious of viral-like symptoms | high | |
| Shortness_of_Breath | suspicious of viral-like symptoms | high | |
| Coughing_up_blood | suspicious of viral-like symptoms | medium | |
| Nausea_or_vomiting | suspicious of viral-like symptoms | medium | |
| Fatigue | suspicious of viral-like symptoms | medium | Fatigue combined with respiratory symptoms make high |
| Sinus_Pain | suspicious of viral-like symptoms | medium | |
| Sore_throat | suspicious of viral-like symptoms | medium | |
| Chills | suspicious of viral-like symptoms | medium | |
| Phlegm | suspicious of viral-like symptoms | low | Make Low |
| Bone_or_joint_pain | suspicious of viral-like symptoms | low | Make low |
| Diarrhea | suspicious of viral-like symptoms | low | Make low if only one day |
| Stuffy_nose | suspicious of viral-like symptoms | low | |
| Loss_of_appetite | suspicious of viral-like symptoms | low | Loss of appetite with other symptoms increase to medium (except fever, which is high) |
| Any persistent symptom for 2 or 3 days | suspicious of viral-like symptoms | low | |
| Stuffy nose AND Swollen eyes AND Fever | suspicious of viral-like symptoms | high | |
| Feel_Sick | suspicious of viral-like symptoms | low | |
| Headache | suspicious of viral-like symptoms | low | Headache with other symptoms increase to medium (except fever make high) |
| Swollen_eyes | suspicious of viral-like symptoms | low | |
| Any medium or low item with fever | | high | |
| **Conditions that DECREASE suspicion of viral-like symptoms** | **THEN** | **Confidence** | |
| Stuffy_nose AND Swollen_eyes AND NOT Fever | NOT suspicious of viral-like symptoms | low | |

Confidence levels

• High = Condition is *strongly* indicative of risk

• Medium = Condition is *somewhat* indicative of risk

• Low = Condition is *slightly* indicative of increase/decrease of risk

## Forecasting model

The forecasting model was used to associate a label of suspicion for viral illness from the Labeling model to the features extracted from the user's cognitive function assessment and physiological signals. The physiological features include (1) single day and (2) rolling averages over 28 days of the heart rate, heart rate variability, respiration rate, activity, sleep latency, sleep duration, composition (light, REM, deep), skin temperature, and sleep efficiency. Physiological features to the exclusion of skin temperature are measured during the night to remove noise due to varying daily activities. The daily cognitive task (PVT) is a sustained-attention, a reaction-timed task that measures the speed with which subjects respond to a visual stimulus [18]. From this data set, the algorithm extracts rules using an information gain-based approach and combines them in a predictive model using a probabilistic graphical network as follows.

The set of probabilistic rules comprises a Markov network. The joint distribution defined by the Markov network can be written as $P(x) = \frac{1}{Z}\exp\left(\sum_j \omega_j f_j(x)\right)$ where $x = (x_1, x_2, \ldots, x_n, y)$ denotes a set of $n+1$ binary variables, out of which the first $n$ are input variables, and $y$ is the output variable. Here, $f_j(x) \in \{1, 0\}$ is a Boolean function corresponding to the rule, $\omega$ is a factor associated with the corresponding rule, $Z$ is the normalization constant. In the current implementation, the relation between the rule's factor $\omega$ and the weight $\psi$ used in the supplementary materials is given by $\psi = \frac{\exp(\omega)}{1+\exp(\omega)}$. More details on the fundamentals of the probabilistic logic can be found in [21–23]. With the joined distribution defined above, the model prediction $s$ for every observation vector $r = (r_1, r_2, \ldots, r_n)$ is computed as the conditional probability of the output variable $y$ as $s = P(y = 1|r)$.

If a training set is available, the model's parameters can be determined by the calibration process, which minimizes the prediction error. Suppose, for the $i$-th training example the model's prediction is $s_i$, and the observed (ground truth) output is $y_i$. We define the cross-entropy loss function as

$$L = -\sum_i y_i \log(s_i) + (1 - y_i)\log(1 - s_i)$$

The calibration process uses the steepest gradient descent to find a combination of rules weights which minimizes the loss function. In our particular implementation we used Limited-memory Broyden–Fletcher–Goldfarb–Shanno algorithm (L-BFGS).

Our model was developed on Stratyfy's Probabilistic Rule Engine, a commercial machine learning platform [26] (see S3 File). In the application to our study, this general framework for creating a rule-based predictive model was applied as follows: In a preprocessing step, the data from the wearable device (e.g., heart rate, temperature, etc.) and the information for the mobile app (e.g., symptoms, results of the PVT, etc.) were collected, checked for completeness, and engineered variables were extracted. We found that, for our study, large gaps in the data had a significant negative impact on the predictive power of the model and, therefore, our efforts were concentrated on cases where most of the required information was actually available. We identified a number of engineered variables (for instance, a ratio of heart rate to heart rate variability) which helped significantly improve the model's predictive power. In order to be used with probabilistic rules, continuous variables are discretized, and then discretized and categorical variables are converted into binary variables by one-hot encoding. The labeling model described above was used to construct the binary output variable, marking for each case days of potential onset of a viral infection. At this point, the setup fit into the context of a standard supervised learning problem: We needed to train a classifier to predict the onset of a disease based on the information available before the actual onset. We opted for the rule-based system described above for several reasons. A main reason was the transparency and interpretability

of our model. In this case, our rule-based system produced models that were fairly small in size (20–50 rules) and still highly accurate. We compared our approach to standard approaches, for example gradient boosting, and found the rule-based approach most promising. Note that, in this study, the rule-based models were used in two ways. In the labelling model, the rules, together with the confidences, were developed and specified by clinical experts. To create the forecasting model, the rules were extracted from the available data via rule mining. For this purpose, we used the Association Rule Mining algorithm [27], which is based on the co-occurrences frequency analysis. After extracting the rules, the weights of the rules were determined by the calibration process outlined above.

## Validation of the model

Model performance was tested with K-fold cross-validation with in our case we perform four rounds of validation (K = 4). One round of cross-validation involves portioning the dataset into complementary subsets, performing the training on one subset and the validation on the other. To reduce variability, multiple rounds of cross-validation are performed using different partitions, and the validation results are combined (averaged) over the rounds to give an estimate of the model's predictive performance. The entire dataset is divided 4 times as 75% for training and 25% for validating the model. The results are then average across the 4 runs of training-validation. The model weights in the final model are obtained by using training dataset of the model. We measure the model's performances at various threshold settings. We also used the area under the curve (AUC) of the receiver operating characteristic (ROC) curve as a threshold-invariant performance measure. Additionally, we report the model's learning performances, i.e., how much data is required to reach the stability of the model. Learning is achieved when adding more data does not significantly impact the performance of the model.

## Results

We enrolled 867 subjects in the study between Apr 7th, 2020, and Aug 1st, 2020 (age ranged from 20 to 76 years old) (Table 2). The data set includes 75,292 unique data points (median number of days of data per participant is 90 days) (see S2 File). 33% (289) unique participants were labeled (via the labeling model) as having symptoms consistent with a viral illness. The forecasting model's inclusion criteria require at least three days of continuous data with no more than one feature missing due to compliance (Fig 1). Of the 767 participants that met the criteria, 276 had missing data for the wearable and 376 for the cognitive assessment. The remaining 115 participants were used to label the wearable and cognitive data as input for the three-day forecasting model. Each day of data was adjudicated by the labeling model, which predicted a 10% occurrence of symptoms consistent with a viral-like illness. The remaining days were labeled as negative or non-suspicious of viral-like illness. From the training dataset,

**Table 2. Study populations demographics.**

| Group | WVU (N = 698) | Vanderbilt (N = 97) | TJU (N = 69) | All (N = 867) |
|---|---|---|---|---|
| Sex–no (%) | | | | |
| Male | 212 (30.3) | 14 (14.4) | 10 (14.5) | 236 (27.2) |
| Female | 252 (36.1) | 40 (41.2) | 21 (30.4) | 313 (36.1) |
| Did Not Respond | 234 (33.5) | 43 (44.3) | 38 (55.1) | 318 (36.7) |
| Age (mean ± SD) | 37.6 ± 11.6 | 37.8 ± 9.7 | 32.8 ± 10.0 | 37.6 ± 11.3 |
| Diabetes Yes-No (%) | 11 (1.6) | 1 (1) | 1 (1.4) | 13 (1.5) |
| Hypertension Yes-No (%) | 31 (4.4) | 7 (7.2) | 2 (2.9) | 40 (4.6) |

the algorithm identified 45 probabilistic rules. These are combined to form the forecasting model (Table 3). The rules contributing to the high probability of developing symptoms within three days are related to low HRV, slower response time to cognitive testing, longer latency to get asleep combined with an increased REM sleep time, and an increased HR. The rules that contribute to a lower probability of developing symptoms are related to lower HR, increased HRV, increased sleep quality, and faster response rate to cognitive testing. Fig 2 provides the model performance as a function of the threshold. Fig 3 illustrates that the model reaches a plateau after about 1500 samples, and that much accuracy cannot be gained by adding more samples. Table 4 reports the precision, recall, and accuracy metrics obtained with a threshold = 0.1 for models with and without cognitive assessment data. The threshold was selected to maximize the balance between precision and recall. The overall accuracy of the model is 82%. The recall positive defined as the true positives (TP) over the total number of positive values (TP/(TP+FN)) is 79% (no PVT:67%). The accuracy of calling negatives (recall negative) defined as the true negatives (TN) over the total amount of negative values (TN/(TN+FP)) is 83% (no PVT:84%). AUC is 89% (no PVT: 83%).

## Discussion

In this study, we measure daily changes in autonomic activity using a wearable device and cognitive assessments via a mobile app. Using machine-learning analytics, we then forecast the onset of symptoms consistent with a viral illness. Specifically, we describe our strategy of using an AI model in conjunction with a non-invasive and readily available technology, which predicts the likelihood of developing symptoms consistent with a viral infection three days before symptom onset with an accuracy of 82%. The model has a false positive rate of 21% (meaning the system would label a non-infected participant as suspicious) and a false-negative of 17% (meaning the system would not detect a suspicious participant). Due to the occurrence of disease in the population, our dataset is unbalanced with more negatives than positives to a ratio of about 4 to 1. The model would detect 79% of individuals who will develop symptoms (i.e., sensitivity) and correctly predicts, almost all of the time (97%, negative predictive value), individuals who will not develop viral-like illness symptoms in the next three days. Conversely, the model precision is 34%. That precision is defined as the ratio of true positives (TP) over positives (P). In other words, if the model flags someone to develop viral-like symptoms in the next three days, the model is correct 34% of the time. Finally, the very little difference in AUCs between each fold suggest that the model is consistently generalizable.

The current model parameters were chosen to provide a conservative framework that warns potentially pre-symptomatic individuals to socially isolate while minimizing warnings to individuals with a low likelihood of developing viral-like symptoms in the next three days. The individuals predicted to be positive (true or false positives) would undergo additional screening and precautions. This framework can be applied as a digital decision-making management tool for public health safety in addition to conventional infection-control strategies.

Other investigators have confirmed the relationship between autonomic activity and the inflammatory response [28–30]. This study suggests a time-dependent relationship between autonomic and cognitive activity and the forecasting of symptoms consistent with a viral illness. We observed consistent changes in the autonomic nervous system function preceding the onset of symptoms. Specifically, differences were observed in HRV, HR, and sleep indices three days before symptom onset. Importantly, this period corresponds to the pre-symptomatic phase of some viral illness such as COVID-19 that is estimated to be 2.5 days [1–4]. In addition to the autonomic changes measured by the wearables, our analyses demonstrate the

**Table 3. Algorithm-derived rules list of 45 rules extracted by the algorithm and used in the model with their relative weights.**

| IF | | THEN | | |
|----|----|----|----|----|
| IF | lbl_score $>= 0.5$ | THEN | suspicious | 0.97 |
| IF | lbl_score in [0.2.. 0.5) | THEN | suspicious | 0.95 |
| IF | (HRV in (30.. 43] AND lbl_score $< 0.2$) | THEN | suspicious | 0.91 |
| IF | (Breath_Average $<= 14.5$ AND MedianResponseTime_AM $> 365$) | THEN | suspicious | 0.90 |
| IF | (Age in (27.. 33] AND lbl_score $< 0.2$) | THEN | suspicious | 0.87 |
| IF | (Sex = Female AND lbl_score $>= 0.5$) | THEN | suspicious | 0.84 |
| IF | (Onset_Latency $> 0.0417$ AND REM $> 1.62$) | THEN | suspicious | 0.83 |
| IF | (Age in (27.. 33] AND lbl_score_t1 $< 0.2$) | THEN | suspicious | 0.78 |
| IF | (Breath_Average $<= 14.5$ AND HRV $> 43$) | THEN | suspicious | 0.77 |
| IF | (HR_delta $<= -1.47$ AND Light $<= 3.42$) | THEN | suspicious | 0.75 |
| IF | (Age $> 46$ AND Sex = Female) | THEN | suspicious | 0.75 |
| IF | (MedianResponseTime_AM in (322.. 365] AND lbl_score $< 0.2$) | THEN | suspicious | 0.73 |
| IF | (Onset_Latency in (0.00333.. 0.0417] AND Sleep_Score $<= 72$) | THEN | suspicious | 0.73 |
| IF | (Onset_Latency in (0.00333.. 0.0417] AND HRV_delta_t1 $<= -4.11$) | THEN | suspicious | 0.73 |
| IF | (AM_Readiness $<= 5.28$ AND Sex = Female) | THEN | suspicious | 0.72 |
| IF | (HR_delta in (-1.47.. 1.38] AND Sex = Male) | THEN | suspicious | 0.70 |
| IF | (E1 $<= 0.274$ AND HRV_base in (30.1.. 43.5]) | THEN | suspicious | 0.69 |
| IF | (MedianResponseTime_PM in (326.. 375] AND Sex = Male) | THEN | suspicious | 0.64 |
| IF | (Onset_Latency in (0.00333.. 0.0417] AND lbl_score in [0.2.. 0.5)) | THEN | suspicious | 0.58 |
| IF | True | THEN | suspicious | 0.11 |
| IF | (HR_Lowest in (55.. 61] AND TLX_Stress_Score in (88.. 163]) | THEN | not suspicious | 0.87 |
| IF | (Age in (27.. 33] AND Sex = Female) | THEN | not suspicious | 0.86 |
| IF | (Light $> 4.31$ AND MedianResponseTime_PM $> 375$) | THEN | not suspicious | 0.86 |
| IF | (HRV in (30.. 43] AND lbl_score_t1 $< 0.2$) | THEN | not suspicious | 0.84 |
| IF | (HR_delta_t1 in (-1.45.. 1.35] AND REM $> 1.62$) | THEN | not suspicious | 0.83 |
| IF | (Sleep_Score $> 82$ AND HRV_delta $> 2$) | THEN | not suspicious | 0.83 |
| IF | (MedianResponseTime_PM in (326.. 375] AND Sleep_Score $<= 72$) | THEN | not suspicious | 0.81 |
| IF | (MedianResponseTime_AM in (322.. 365] AND lbl_score_t1 $< 0.2$) | THEN | not suspicious | 0.80 |
| IF | (E4_t2 in (1.44.. 2.32] AND Score_Efficiency in (83.. 96]) | THEN | not suspicious | 0.79 |
| IF | (E1_t2 $<= 0.27$ AND Light $<= 3.42$) | THEN | not suspicious | 0.78 |
| IF | (HR_Lowest $> 61$ AND HRV in (30.. 43]) | THEN | not suspicious | 0.77 |
| IF | (E5_t1 in (-0.753.. 0.724] AND HR $<= 62$) | THEN | not suspicious | 0.77 |
| IF | (E1 $<= 0.274$ AND HR_delta in (-1.47.. 1.38]) | THEN | not suspicious | 0.77 |
| IF | (Onset_Latency $> 0.0417$ AND HRV_delta_t1 $> 1.99$) | THEN | not suspicious | 0.76 |
| IF | (E5 $> 0.758$ AND HRV_delta_t1 $<= -4.11$) | THEN | not suspicious | 0.75 |
| IF | (Breath_Average $<= 14.5$ AND Sex = Female) | THEN | not suspicious | 0.74 |
| IF | (E5_t2 in (-0.74.. 0.779] AND Temperature_Delta in (-0.1.. 0.08]) | THEN | not suspicious | 0.74 |
| IF | (HR_delta_t2 in (-1.45.. 1.43] AND Temperature $<= 97.6$) | THEN | not suspicious | 0.73 |
| IF | (Duration_Integer_hr $<= 5.8$ AND E1 $<= 0.274$) | THEN | not suspicious | 0.73 |
| IF | (E4_t1 in (1.45.. 2.3] AND E5_t1 in (-0.753.. 0.724]) | THEN | not suspicious | 0.71 |
| IF | (TLX_Stress_Score $> 163$ AND HRV in (30.. 43]) | THEN | not suspicious | 0.68 |
| IF | (Light in (3.42.. 4.31] AND Sex = Male) | THEN | not suspicious | 0.67 |
| IF | (Age in (33.. 37.5] AND TLX_Stress_Score $<= 88$) | THEN | not suspicious | 0.66 |
| IF | (Sex = Male AND TLX_Stress_Score $> 163$) | THEN | not suspicious | 0.63 |
| IF | Age in (27.. 33] | THEN | not suspicious | 0.61 |

Rules are aggregated to forecast suspicion of a viral disease in a participant.

**A.**

**B.**

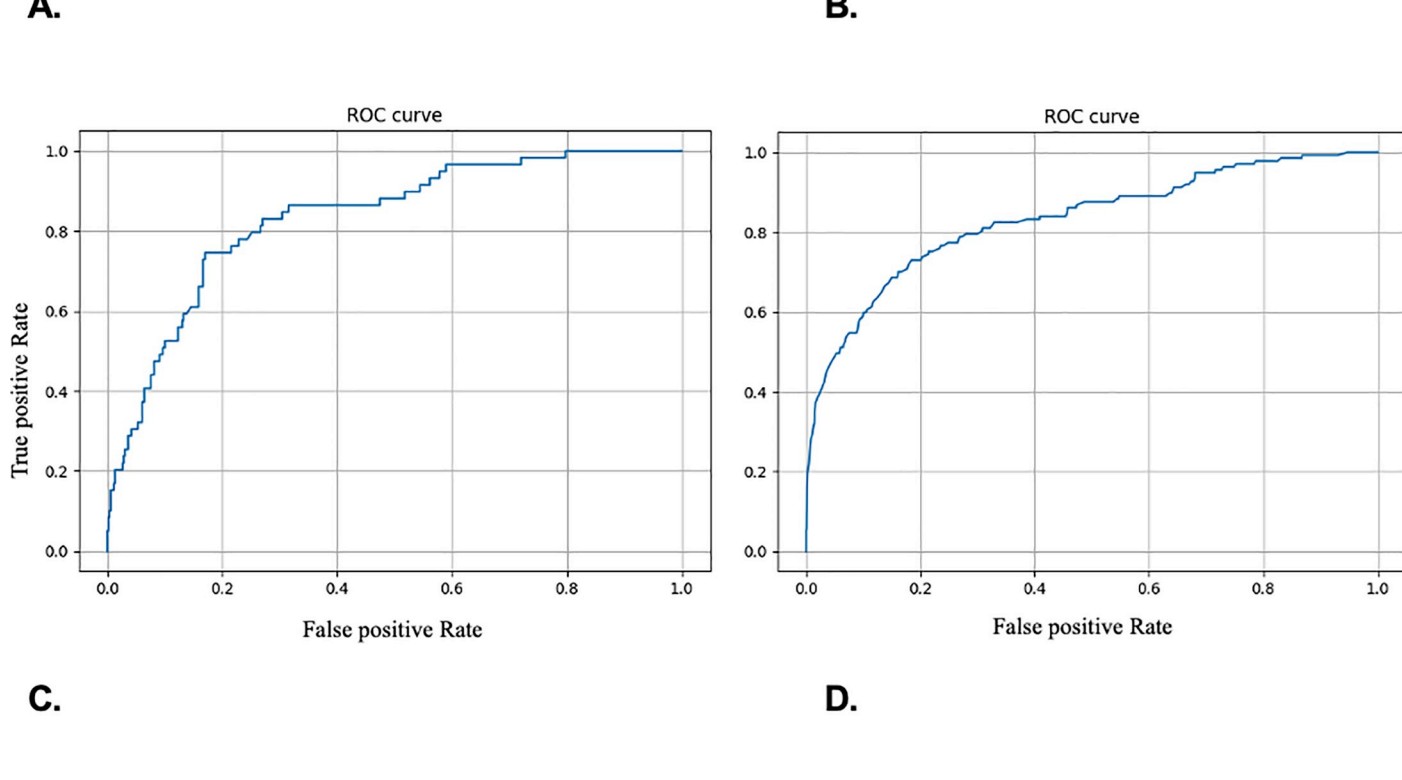

**C.**

**D.**

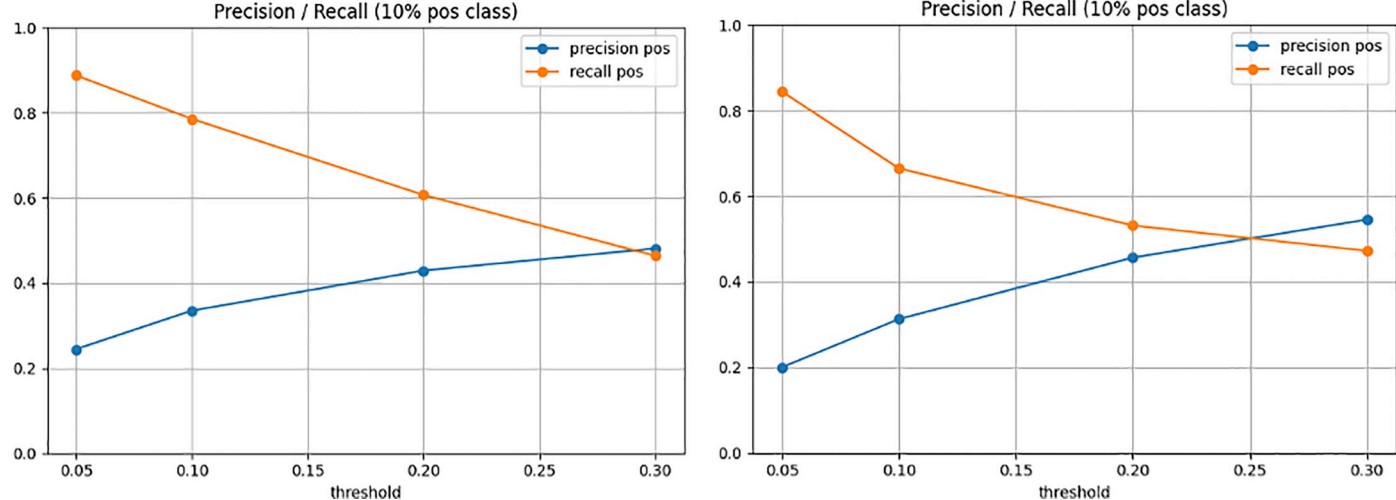

**Fig 2.** ROC (False Positive Rate vs True Positive Rate) and Precision/recall curves for the forecasting model with (A, C) and without cognitive assessment (B, D).

additional value of cognitive assessments (PVT) to predict symptoms consistent with a viral illness.

There are several limitations to this study. First, we did not diagnose infection nor measure infection markers in each individual. Instead, we relied on self-reported symptoms known to be associated with the occurrence of a viral infection. Without definitive diagnostics, we cannot confirm the presence of viral infection among persons who self-report symptoms. In the next phase of the study, we plan to test specific viruses (e.g., influenza and SARS CoV-2). The

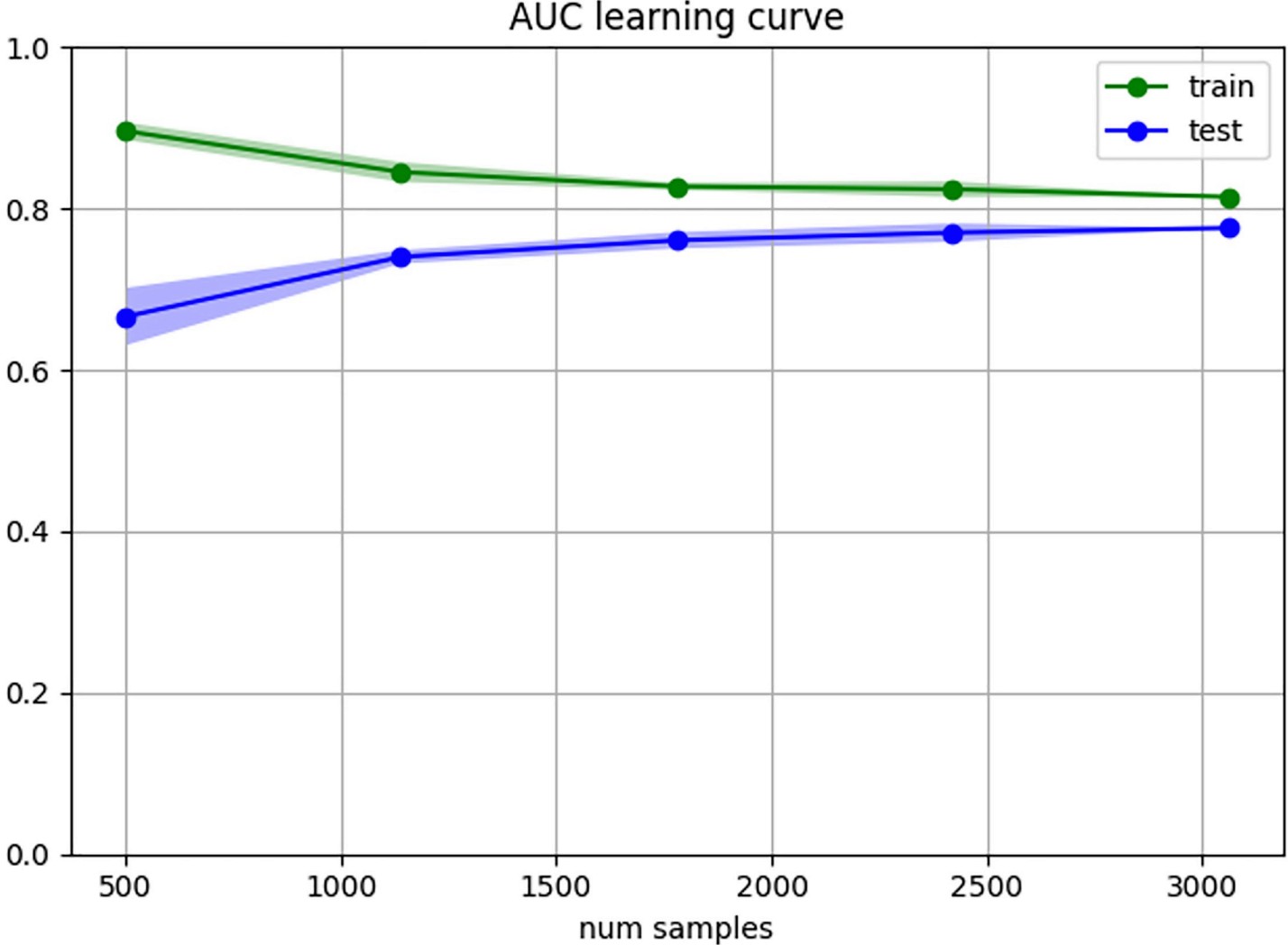

**Fig 3. Model performance (learning and stability).** The shade colors represent the variance of results created by the K-Fold(4) validation.

participants in this study are limited to front-line health care workers. Our model would benefit from being extended to other populations. Finally, participant compliance to consistently use their wearable and the app remains a challenge. Non-compliance among our participants reduced the usable data set. We plan on developing additional models to impute the data in an efficient way in order to extend the usability of the forecasting model. While we have demonstrated that the dataset is sufficient to reach this model's predictive stability, additional data will provide further insights and reinforce the conclusions.

Viral infections have physical, cognitive, behavioral, and environmental influences and stressors that impact infection risk [31, 32]. To our knowledge, this is the first study using wearables and apps with machine learning to predict symptoms consistent with viral infection three days before their onset. The demonstrated approach to forecasting the onset of viral illness-like symptoms offers a novel digital decision-making tool for public health safety by potentially limiting viral transmission.

**Table 4. Model performance with and without PVT.** A. With PVT. B. Without PVT.

| A | | | | |
|---|---|---|---|---|
| Threshold | 0.05 | 0.1 | 0.2 | 0.3 |
| Recall Pos | 0.89 | 0.79 | 0.61 | 0.46 |
| Recall Neg | 0.70 | 0.83 | 0.91 | 0.94 |
| Precision Pos | 0.24 | 0.34 | 0.43 | 0.47 |
| Precision Neg | 0.98 | 0.97 | 0.95 | 0.94 |
| Accuracy | 0.72 | 0.82 | 0.88 | 0.90 |
| AUC | 0.88 | | | |
| B | | | | |
| Threshold | 0.05 | 0.1 | 0.2 | 0.3 |
| Recall Pos | 0.8 | 0.66 | 0.53 | 0.47 |
| Recall Neg | 0.62 | 0.83 | 0.92 | 0.95 |
| Precision Pos | 0.20 | 0.31 | 0.45 | 0.54 |
| Precision Neg | 0.97 | 0.95 | 0.94 | 0.94 |
| Accuracy | 0.64 | 0.82 | 0.88 | 0.91 |
| AUC | 0.83 | | | |

## Supporting information

**S1 Table. Data dictionary.** Data dictionary of each data element used in the model.
(PDF)

**S2 Table. Disease onset model rules.** The following table reports the probabilistic weights for each rule of the symptom onset forecasting model.
(PDF)

**S1 File. List of questions.** List of questions asked to the participants.
(PDF)

**S2 File. Dataset and inclusion/exclusion criteria.** Inclusion/exclusion criteria and description of data set.
(PDF)

**S3 File. Probabilistic rule engine.** Detailed description of the Probabilistic Rule Engine.
(PDF)

**S4 File. Validation approach.** Detailed description of the validation approach.
(PDF)

**S5 File. Labeling model.** Detailed description of the labeling model.
(PDF)

## Acknowledgments

We would like to thank the teams and clinical coordinators from Vanderbilt University and Thomas Jefferson University, who have provided the support to recruit and support the participants. We would like to thank the OURARing team for their partnership and integration of their data system.

## Author Contributions

**Conceptualization:** Pierre-François D'Haese, Victor Finomore, Sally Hodder, Ali R. Rezai.

**Data curation:** Pierre-François D'Haese, Dmitry Lesnik, Tobias Schaefer.

**Formal analysis:** Pierre-François D'Haese, Victor Finomore, Dmitry Lesnik, Tobias Schaefer, Sally Hodder.

**Funding acquisition:** Victor Finomore, Ali R. Rezai.

**Investigation:** Pierre-François D'Haese, Clay Marsh, Ali R. Rezai.

**Methodology:** Pierre-François D'Haese, Victor Finomore, Tobias Schaefer, Sally Hodder, Clay Marsh, Ali R. Rezai.

**Project administration:** Pierre-François D'Haese, Peter E. Konrad, Sally Hodder, Ali R. Rezai.

**Resources:** Pierre-François D'Haese, Victor Finomore, Peter E. Konrad, Clay Marsh, Ali R. Rezai.

**Software:** Pierre-François D'Haese, Dmitry Lesnik.

**Supervision:** Pierre-François D'Haese, Victor Finomore, Laura Kornhauser, Peter E. Konrad, Sally Hodder, Clay Marsh, Ali R. Rezai.

**Validation:** Pierre-François D'Haese, Victor Finomore, Peter E. Konrad, Ali R. Rezai.

**Visualization:** Pierre-François D'Haese, Victor Finomore, Ali R. Rezai.

**Writing – original draft:** Pierre-François D'Haese, Victor Finomore, Sally Hodder.

**Writing – review & editing:** Pierre-François D'Haese, Victor Finomore, Peter E. Konrad, Sally Hodder, Clay Marsh, Ali R. Rezai.

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
