## [Decision Letter · Decision Letter 0]

20 Jan 2021

PONE-D-20-36216

Prediction of viral symptoms using wearable technology and artificial intelligence: A pilot study in healthcare workers

PLOS ONE

Dear Dr. DHAESE,

Thank you for submitting your manuscript to PLOS ONE. After careful consideration, we feel that it has merit but does not fully meet PLOS ONE’s publication criteria as it currently stands. Therefore, we invite you to submit a revised version of the manuscript that addresses the points raised during the review process.

We look forward to receiving your revised manuscript.

Kind regards,

Nizam Uddin Ahamed, PhD

Academic Editor

PLOS ONE

Journal Requirements:

"The authors have declared that no competing interests exist"

We note that one or more of the authors are employed by a commercial company: Stratyfy, Inc.

2.1. Please provide an amended Funding Statement declaring this commercial affiliation, as well as a statement regarding the Role of Funders in your study. If the funding organization did not play a role in the study design, data collection and analysis, decision to publish, or preparation of the manuscript and only provided financial support in the form of authors' salaries and/or research materials, please review your statements relating to the author contributions, and ensure you have specifically and accurately indicated the role(s) that these authors had in your study. You can update author roles in the Author Contributions section of the online submission form.

2.2. Please also provide an updated Competing Interests Statement declaring this commercial affiliation along with any other relevant declarations relating to employment, consultancy, patents, products in development, or marketed products, etc.  

4. Please include your tables as part of your main manuscript and remove the individual files. Please note that supplementary tables should be uploaded as separate "supporting information" files.

Reviewers' comments:

Reviewer's Responses to Questions

**Comments to the Author**

1. Is the manuscript technically sound, and do the data support the conclusions?

Reviewer #1: Partly

Reviewer #2: Yes

2. Has the statistical analysis been performed appropriately and rigorously? 

Reviewer #1: No

Reviewer #2: Yes

3. Have the authors made all data underlying the findings in their manuscript fully available?

Reviewer #1: No

Reviewer #2: No

4. Is the manuscript presented in an intelligible fashion and written in standard English?

Reviewer #1: Yes

Reviewer #2: Yes

5. Review Comments to the Author

Reviewer #1: This is an incredibly timely study tackling the most urgent issue facing us all right now. It presents a wearables-based approach for capturing physiological and physical behaviours, and using these to predict whether someone may potentially have coronavirus. Importantly, the work aims to predict this even before major symptoms develop. Should it work, this would be a significant tool in helping us to control the spread of this virus.

The study is, for a wearables-based work, extensive - with over 800 frontline workers tracked for around 90 days. Unfortunately, due to data collection issues, only about 115 participants are used in the final results.

Nonetheless, this gives us a preliminary and encouraging result that can be acted upon.

Important though this work is, I have some concerns regarding both the methodology used and the presentation of the results. These should be addressed as speedily as is possible to ensure the work is fit for publication - and, most importantly, replication.

First off, the machine learning component is not clear. The model is described as a probabilistic graphical network acting on a combination of binary features. The exact operation or preprocessing applied to these features, or indeed what these features are, is not specified in the main paper. Further information is supplied in the supplementary material, however it would enhance reader understanding considerably to give some clearer example on, for example, how measured heart rate is incorporated into the model. (Even in the supplementary material, the model description could be improved.) This whole section should be written in a clearer, fuller, way using appropriate terminology to allow easier replication. It would also be helpful to know why this model was chosen over other potential machine learning approaches.

The training and testing procedure of the algorithm is also unclear. The supplementary material goes some way to clarifying this, but the main text could be tightened up somewhat. As K-fold cross-validation is used, it would help to directly specify the value of K (=4 in this case). Specifying a 25% test set split is not quite enough because this could imply a single leave-one test set out evaluation. With the results being averaged over the K folds, then some additional measure of variance might be helpful. Following from this, the supplementary material provides a table of model weights - how are these arrived at from the K-folds?

Finally, the results presentation should be improved. The use of several complementary metrics makes sense, however the headline result in the abstract and discussion present misleading figures. The abstract reports a positive recall, or sensitivity, of 97%, when in fact it should be 79%. According to the tables, the negative recall, or specificity, should be around 83%, however the discussion states that "almost all of the time (97%), individuals who will not develop viral-like illness symptoms". There may be additional errors or confusions in the reporting of these results that I have not found, and I would recommend going through these in detail again.

Some specific presentation issues:

- Introduction, para 3: sentence structure on 'Outputs from wearable...health and disease'

- See also 'Besides, subject-reported ... COVID-19'

- Forecasting Model: unclear statement 'where x is a set of binary variables (among which there are n input and one output variable)...' What does n refer to? This whole section needs expanding and clarification.

- Results: 867 -> 767 -> 115. Was it not possible to salvage some of the missing data? For example, include some of the 376 missing cognitive data that had some wearable data towards training the forecasting model?

Reviewer #2: The authors present a rule based probabilistic analysis for determining the onset of symptoms for SARS-CoV-2 among the subjects in the study. They use different wearables to taking measurements of the people involved in their study and try to determine the people susceptible to infection with SARS-CoV-2. The work is interesting and is of value for everyone considering the entire world is grappling with the pandemic.

The concerns that I have about the manuscript in its current state are as follows:

1. The authors state that "The model is calibrated using an accepted method (cross-entropy loss function, see supplementary material) that finds the set of weights , which minimizes the prediction errors using the training model". After checking the supplementary material, I found the authors state that the cross-entropy function maximizes the maximum likelihood estimation (MLE). However, it is not clear to me what is the y in the process to calculate the conditional probability P(y=1|r), the y is not explained clearly in supplementary material (SM). Is it the subjects in the study who were eventually infected by COVID-19? Moreover the cross entropy loss function is Loss = -y*log(y). This term is then summed over all the possible classes in the dataset. How are the rules fit into this framework? If the authors are not using something like this how are weights calculated?

This calculation of weights is a very important detail in the machine learning (ML) model developed by the authors but it is a bit unclear. I would like the authors to explain it in greater detail and write about it in the main manuscript rather than SM.

2. The authors use the exact same line twice in the paper once in abstract and once in discussion

"Conversely, the model correctly predicts as positive, 34% of the time, individuals who will develop viral-like illness symptoms in the next three days. "

Details about the implication of the statement in the discussion section are missing

What does 34% signify in this case, does it mean that the accuracy for viral like symptoms is 34%?

3. Are the rules stated in the forecasting model created by experts after observing the data? If not, how were they created? The rules in Table 6.1 clearly state expert rule used in labelling model but there is no such mention of that in table 7.1. Assuming it was created by experts it would have been more useful to use some ML techniques that figure about the rules themselves like random forests and other decision tree based models.

4.Intuitively, the greater weight would imply greater influence of the rule in predicting the label. However, most of the weights presented in the study are fairly close. For example: Shortness of Breath, Coughing up blood, Sore throat, Chills, Phlegm, Diarrhea have almost the same weight ~0.82. No analysis of why this is happening has been presented, is it because of the nature of the data? some of the parameters presented in the example are distinctly different. Another thing that I found interesting was that persistence of symptoms had lesser weight than cough and fever. However, if the symptoms in the table are persistent for 2-3 days then the person would be more worried. It would be interesting to analyze the weights in greater detail. It will make the study more clearer and accessible

5.This is a minor concern: The very essence of ML lies in learning from the features and developing a mapping between labels (in this case the rules and labels). If the authors use some more advanced ML models in their future studies perhaps there will no need to draft the rules. One problem that I see that the designed rules may not be exhaustive and some of the important rules may be missing from the model in the current state.

Overall, I feel that the work and the methodology are of value great research community. However, the authors have some small gaps in their study and need to address a few things in greater detail and clarity.

6. PLOS authors have the option to publish the peer review history of their article (what does this mean?). If published, this will include your full peer review and any attached files.

Reviewer #1: **Yes: **J. Ward

Reviewer #2: No

---

## [Author Response · Author response to Decision Letter 0]

2 Jun 2021

Response to Reviewers

First, we would like to thank the Reviewers for the detailed reading of our manuscript and all the valuable comments. Please find below a list of changes in response to their comments.

Responses to the comments of the first Reviewer:

1. The Reviewer wrote:

“First off, the machine learning component is not clear. The model is described as a probabilistic graphical network acting on a combination of binary features. The exact operation or preprocessing applied to these features, or indeed what these features are, is not specified in the main paper. Further information is supplied in the supplementary material, however it would enhance reader understanding considerably to give some clearer examples on, for example, how measured heart rate is incorporated into the model. (Even in the supplementary material, the model description could be improved.) This whole section should be written in a clearer, fuller, way using appropriate terminology to allow easier replication. It would also be helpful to know why this model was chosen over other potential machine learning approaches.”

Our reply: 

To address the reviewer’s comments, we rewrote the section about forecasting to clarify our approach. In particular, we incorporated some of the supplementary material into the main text to add more information about the underlying algorithm in this section and the relevant references to Markov Logic. We also added information about the data flow and the preprocessing to facilitate the replication of our approach.

2. The Reviewer wrote:

“The training and testing procedure of the algorithm is also unclear. The supplementary material goes some way to clarifying this, but the main text could be tightened up somewhat. As K-fold cross-validation is used, it would help to directly specify the value of K (=4 in this case). Specifying a 25% test set split is not quite enough because this could imply a single leave-one test set out evaluation. With the results being averaged over the K folds, then some additional measure of variance might be helpful. Following from this, the supplementary material provides a table of model weights - how are these arrived at from the K-folds?”

Our reply:

We made the appropriate changes to the main text; in particular, we specified the value of K=4:

“Model performance was tested with K-fold cross-validation using the reserved 25% portion of the data, i.e. K=4 in our case.”

As usual in supervised learning, the model weights in the final model are not found via averaging but instead obtained by using all available data to train the model. To clarify this in the main text, we added the following sentence:

“The model weights in the final model are obtained by using all available data to train the model.”

The differences in the AUCs of the different folds were too small to be of interest. Moreover, our samples are appropriately stratified, avoiding a single leave-one test set out evaluation. 

3. The Reviewer wrote:

“Finally, the results presentation should be improved. The use of several complementary metrics makes sense; however the headline result in the abstract and discussion present misleading figures. The abstract reports a positive recall, or sensitivity, of 97%, when in fact it should be 79%. According to the tables, the negative recall, or specificity, should be around 83%; however, the discussion states that "almost all of the time (97%), individuals who will not develop viral-like illness symptoms". There may be other errors or confusions in the reporting of these results that I have not found, and I would recommend going through these in detail again.”

Our reply:

We agree with the Reviewer and modified our report of results to increase clarity. The numbers in the abstract refer to Table A, particularly to the column reporting results for the threshold chosen to be 0.1. For this case, the sensitivity is indeed 79%, and the 93% refer to the negative predictive value (TN/(TN+FN)). We made the following changes in the text to improve clarity:

“We describe our strategy using an AI model that can predict, with 82% accuracy (negative predictive value 97%, specificity 83%, sensitivity 79%, precision 34%)”

We also added the words “negative predictive value” and “precision” to the main text:

The model would detect 79% of individuals who will develop symptoms (i.e., sensitivity) and correctly predicts, almost all of the time (97%, negative predictive value), individuals who will not develop viral-like illness symptoms in the next three days. Conversely, the model precision is 34%. Remember that precision is defined as the ratio of true positives (TP) over positives (P). In other words, if the model flags someone to develop viral-like symptoms in the next three days, the model is correct 34% of the time. 

4. The Reviewer wrote:

“Some specific presentation issues:

 - Introduction, para 3: sentence structure on 'Outputs from wearable...health and disease'

 - See also 'Besides, subject-reported ... COVID-19'”

Our reply:

We changed the sentences that the reviewer pointed out. The new version reads:

Outputs from wearable technology, including heart rate (HR), heart rate variability (HRV), respiration rate (RR), temperature, blood oxygenation, sleep, and other physiological assessments, are increasingly being explored in studies of health and disease 10–12. Moreover, various subject-reported symptoms captured on mobile apps transform both surveillance and contact tracing management strategies for COVID-19 13–15.

5. The Reviewer wrote:

“Forecasting Model: unclear statement 'where x is a set of binary variables (among which there are n input and one output variable)...' What does n refer to? This whole section needs expanding and clarification.”

Our reply:

We rewrote this section to address the Reviewer’s comments. We rephrased the paragraph as follows:

The joint distribution defined by the Markov network can be written as where denotes a set of n+1 binary variables, out of which the first n are input variables, and y is the output variable. Here, is a Boolean function corresponding to the rule, is a factor associated with the corresponding rule, is the normalization constant.

6. The Reviewer wrote:

“Results: 867 -> 767 -> 115. Was it not possible to salvage some of the missing data? For example, include some of the 376 missing cognitive data that had some wearable data towards training the forecasting model?”

Our reply:

In principle, we agree with the Reviewer that some of this data could be attributed appropriately to train the forecasting model. We did some research in this direction, but the preliminary results based on naïve imputation techniques were not very encouraging such that more research is necessary in order to salvage the missing data. In order to capture this in the text, we added the following sentence in the discussion:

We plan to develop additional models to attribute the data efficiently to extend the forecasting model’s usability.

Responses to the comments of the second Reviewer:

1. The Reviewer wrote:

“1. The authors state that "The model is calibrated using an accepted method (cross-entropy loss function, see supplementary material) that finds the set of weights , which minimizes the prediction errors using the training model". After checking the supplementary material, I found the authors state that the cross-entropy function maximizes the maximum likelihood estimation (MLE). However, it is not clear to me what is the y in the process to calculate the conditional probability P(y=1|r), the y is not explained clearly in supplementary material (SM). Is it the subjects in the study who were eventually infected by COVID-19? Moreover the cross entropy loss function is Loss = -y*log(y). This term is then summed over all the possible classes in the dataset. How are the rules fit into this framework? If the authors are not using something like this how are weights calculated?

 This calculation of weights is a very important detail in the machine learning (ML) model developed by the authors but it is a bit unclear. I would like the authors to explain it in greater detail and write about it in the main manuscript rather than SM.”

Our reply:

To address the reviewer’s concerns, we rewrote the section on the forecasting model and included more material from the supplementary material in the main text. This also includes relevant references in the main text.

2. The Reviewer wrote:

“2. The authors use the same line twice in the paper, once in abstract and once in the discussion.

 "Conversely, the model correctly predicts as positive, 34% of the time, individuals who will develop viral-like illness symptoms in the next three days. "

 Details about the implication of the statement in the discussion section are missing.

 What does 34% signify in this case? Doesit mean that the accuracy for viral-likesymptoms is 34%?”

Our reply:

To clarify the text, we added the word “precision” to the sentence in the abstract. Precision is defined as the ratio TP/P = TP/(TP+FP) (where TP are the true positives and FP are the false positives). We also made the following changes in the discussion:

Conversely, the model precision is 34%. Remember that precision is defined as the ratio of true positives (TP) over positives (P). In other words, if the model flags someone to develop viral-like symptoms in the next three days, the model is correct 34% of the time. 

3. The Reviewer wrote:

“3. Are the rules stated in the forecasting model created by experts after observing the data? If not, how were they created? The rules in Table 6.1 clearly state expert rule used in labelling model but there is no such mention of that in table 7.1. Assuming it was created by experts it would have been more useful to use some ML techniques that figure about the rules themselves like random forests and other decision tree based models.”

Our reply:

The rules used in the forecasting model are found via rule mining from our algorithm. The rewritten section on forecasting now contains more information to clarify this point. In particular:

To create the forecasting model, however, the rules were extracted from the available data via rule mining and the weights of the rules were determined by the calibration process.

Also, note that we were focusing on developing an interpretable rule-based model consisting of a relatively small number of rules in our approach. Random forest usually produces a large number of rules, which makes this approach much less interpretable. In our research, the probabilistic rules yielded the most promising results when developing an accurate yet transparent model.

4. The Reviewer wrote:

“4.Intuitively, the greater weight would imply greater influence of the rule in predicting the label. However, most of the weights presented in the study are fairly close. For example: Shortness of Breath, Coughing up blood, Sore throat, Chills, Phlegm, Diarrhea have almost the same weight ~0.82. No analysis of why this is happening has been presented, is it because of the nature of the data? some of the parameters presented in the example are distinctly different. Another thing that I found interesting was that persistence of symptoms had lesser weight than cough and fever. However, if the stable’s symptoms are persistent for 2-3 days, the person would be more worried. It would be interesting to analyze the weights in greater detail. It will make the study clearer and more accessible.”

Our reply:

The rules and weights of the labeling were dictated from the input of medical and epidemiologists experts. This is different from the forecasting model in which the rules are found via rule mining by our algorithm and the weights set through algorithm calibration. To clarify the origin of the rules and weights of the labeling model, we added the following sentences:

The rules and weights in this model were found from expert input and reviewed by our medical team. In particular, the weights associated with the rules were chosen to minimize the labeling error assessed by medical experts. We also fine-tuned some rule weights by fitting the model to a small synthetic data set containing a few most typical symptom combinations. The actual calculation of the labeling model’s output score happens by applying the probabilistic logic machinery, described in more detail in the next paragraph.

We agree with the Reviewer that the labeling model itself, together with the rules and weights, would warrant a broader discussion. In this paper, however, the labeling model is mainly used to define the output variable and thus set the forecasting model stage, which is in this paper’s focus. However, we plan to extend the discussion with our medical team and look into the weights more in-depth to publish insights focusing on the labeling model itself.

5. The Reviewer wrote:

“5.This is a minor concern: The very essence of ML lies in learning from the features and developing a mapping between labels (in this case the rules and labels). If the authors use some more advanced ML models in their future studies perhaps there will no need to draft the rules. One problem that I see that the designed rules may not be exhaustive and some of the important rules may be missing from the model in the current state.”

Our reply: 

We assume that the Reviewer’s comments are in regard to the labeling model. The Reviewer is correct in the sense that rule-based systems are most powerful when there is a combination of expert input and rule-mining from the data. In the particular case of the labeling model in this study, the output variable needed to be defined properly and the idea was to model this in the same way a doctor would define the onset of a viral infection looking at symptoms present. The Reviewer is correct in the sense that, with more data and more input, this definition of onset might be refined by data-driven rules in the future.

---

## [Decision Letter · Decision Letter 1]

7 Jul 2021

PONE-D-20-36216R1

Prediction of viral symptoms using wearable technology and artificial intelligence: A pilot study in healthcare workers

PLOS ONE

Dear Dr. DHAESE,

Thank you for submitting your manuscript to PLOS ONE. After careful consideration, we feel that it has merit but does not fully meet PLOS ONE’s publication criteria as it currently stands. Therefore, we invite you to submit a revised version of the manuscript that addresses the points raised during the review process.

We look forward to receiving your revised manuscript.

Kind regards,

Nizam Uddin Ahamed, PhD

Academic Editor

PLOS ONE

Journal Requirements:

Reviewers' comments:

Reviewer's Responses to Questions

**Comments to the Author**

1. If the authors have adequately addressed your comments raised in a previous round of review and you feel that this manuscript is now acceptable for publication, you may indicate that here to bypass the “Comments to the Author” section, enter your conflict of interest statement in the “Confidential to Editor” section, and submit your "Accept" recommendation.

Reviewer #1: (No Response)

Reviewer #2: All comments have been addressed

2. Is the manuscript technically sound, and do the data support the conclusions?

Reviewer #1: Partly

Reviewer #2: Yes

3. Has the statistical analysis been performed appropriately and rigorously? 

Reviewer #1: Yes

Reviewer #2: Yes

4. Have the authors made all data underlying the findings in their manuscript fully available?

Reviewer #1: No

Reviewer #2: Yes

5. Is the manuscript presented in an intelligible fashion and written in standard English?

Reviewer #1: Yes

Reviewer #2: Yes

6. Review Comments to the Author

Reviewer #1: Following from the earlier review, this paper is both timely and extremely relevant. Frustratingly, the initial round of revisions did not fully address all of the concerns raised and so I would suggest a final round of  revisions focusing on clarity and reproducibility of the work.

The cross-validation approach described in the main text remains unclear. The revision now states that "Model performance was tested with K-fold cross-validation using the reserved 25% portion of the data, i.e. K=4 in our case.”  This reads as if K-fold CV is being applied to the 25% (the reserved test set), rather than the training set as would be expected. This should just be a matter of clarification in the text. 

The fact that there was very little difference in the AUCs for each fold is  a good thing, perhaps worth mentioning as it suggests that the model is consistently generalisable. 

Rule mining is stated as the main method used to build the forecasting model, however very little detail is given as to what rule mining actually is, or how exactly it is implemented. There isn't even a reference given on the technique. Some further detail on this would be appreciated.

The revised statement about model weights is concerning. The use of 'all available data' would suggest that there is no separation of training and test data for evaluating the final model. Just to be clear, and avoid any suspicion of overfitting, can you clarify that this model is only then applied to previously unseen data?

The calibration process uses gradient descent, however there remains a lack of clarity on the exact implementation used or the various design choices made. The one-line description in the main text is  vague and includes the statement,  'or any of its variants', which is too unspecific for a reproducible work. It would aid the reader to include further details on the implementation. The commercial implementation that is used should also be mentioned directly in the main text (rather than simply referenced). Ideally some implementation details on this could be included in the supplementary text, too. Considering that the commercial company which implemented this is included as an affiliate on the paper, it does not seem unreasonable to expect a bit more detail. 

Supplementary material. There is a disconnect between some of the text and the provided tables and figures. Generally, all tables need to be clearly referenced (and include some caption). For example, p12 states, "the learning curve presented in the table,"... yet does not specify which table.

Further comments:

- The cross-entropy function does not render well on the PDF that this reviewer received - several variables were replaced by black rectangles.

- The references all link to a paperpile.com repository which is not accessible (to this reviewer at least)

- The ROC plots need labels on the axes (e.g. Fig2)

- All tables should include descriptive captions

- Figure 3 - please specify in the caption what the variance represents

Reviewer #2: The authors have improved the manuscript significantly. The clarity of the manuscript has been significantly enhanced.

I would like the authors to add one final detail in the manuscript .

1. What is the distribution of the labels in data used to train the model. Since, a lot of data is missing due non-compliance I think it is a relevant detail that needs to be added. For eg: If the dataset is skewed towards one class the numbers indicating the performance can be sometimes misleading. As a constant model that predicts a single class all the time can also have high scores on performance metrics. Addition of this data will help increase the confidence on this developed model

7. PLOS authors have the option to publish the peer review history of their article (what does this mean?). If published, this will include your full peer review and any attached files.

Reviewer #1: **Yes: **J Ward

Reviewer #2: No

---

## [Author Response · Author response to Decision Letter 1]

12 Aug 2021

First, we would like to thank the Reviewers for the detailed reading of our manuscript and all the valuable comments. Please find below a list of changes in response to their comments.

Responses to the comments of the first Reviewer:

 The Reviewer wrote:

“Following from the earlier review, this paper is both timely and extremely relevant. Frustratingly, the initial round of revisions did not fully address all of the concerns raised and so I would suggest a final round of revisions focusing on clarity and reproducibility of the work”

Our reply: We thank the reviewer and agree that these revisions will increase clarity of the 

work. We have revised the manuscript following each review.

 The Reviewer wrote:

“The cross-validation approach described in the main text remains unclear. The revision now states that "Model performance was tested with K-fold cross-validation using the reserved 25% portion of the data, i.e. K=4 in our case.” This reads as if K-fold CV is being applied to the 25% (the reserved test set), rather than the training set as would be expected. This should just be a matter of clarification in the text”

Our reply: We thank the reviewer and agree that these revisions will increase clarity of the 

work. We have revised the manuscript following each review. We have modified the text as 

follows:

Model performance was tested with K-fold cross-validation with in our case we perform four rounds of validation (K=4). One round of cross-validation involves portioning the dataset into complementary subsets, performing the training on one subset and the validation on the other. To reduce variability, multiple rounds of cross-validation are performed using different partitions, and the validation results are combined (averaged) over the rounds to give an estimate of the model's predictive performance. The entire dataset is divided 4 times as 75% for training and 25% for validating the model. The results are then average across the 4 runs of training-validation.

 The Reviewer wrote:

“The fact that there was very little difference in the AUCs for each fold is a good thing, perhaps worth mentioning as it suggests that the model is consistently generalizable. ”

Our reply: We agree with the reviewer that the AUCs variations across folds is a good thing. It is mentioned in the text already in the result section as a reference to figure 3 as “Figure 3 illustrates that the model reaches a plateau after about 1500 samples, and that much accuracy cannot be gained by adding more samples“

We have added a mention of it in the discussion as well as the following statement:

Finally, the very little difference in AUCs between each fold suggest that the model is consistently generalizable.

 The Reviewer wrote:

“Rule mining is stated as the main method used to build the forecasting model, however very little detail is given as to what rule mining actually is, or how exactly it is implemented. There isn't even a reference given on the technique. Some further detail on this would be appreciated.”

Our reply: We agree and we have added some information about rule mining. The rule mining is done using the Assocation Rule Mining Algorithm published by Hipps et al in June 2000. We have added a reference to that paper as well for full reproducibility.

The paragraph added in the Method section now reads:

To create the forecasting model, the rules were extracted from the available data via rule mining. For this purpose, we used the Association Rule Mining algorithm [Jochen Hipp, Ulrich Güntzer, and Gholamreza Nakhaeizadeh. 2000. Algorithms for association rule mining — a general survey and comparison. SIGKDD Explor. Newsl. 2, 1 (June, 2000), 58–64. DOI:https://doi.org/10.1145/360402.360421], which is based on the co-occurrences frequency analysis. After extracting the rules, the weights of the rules were determined by the calibration process outlined above.

And reference 32 was added 

32. Jochen Hipp, Ulrich Güntzer, and Gholamreza Nakhaeizadeh. 2000. Algorithms for association rule mining — a general survey and comparison. SIGKDD Explor. Newsl. 2, 1 (June, 2000), 58–64. DOI:https://doi.org/10.1145/360402.360421

 The Reviewer wrote:

“The revised statement about model weights is concerning. The use of 'all available data' would suggest that there is no separation of training and test data for evaluating the final model. Just to be clear, and avoid any suspicion of overfitting, can you clarify that this model is only then applied to previously unseen data?”

Our reply: We have rewritten the statement to clarify that indeed the model is trained only on the training dataset.

It now reads as: The entire dataset is divided 4 times as 75% for training and 25% for validating the model. The results are then average across the 4 runs of training-validation. The model weights in the final model are obtained by using training dataset of the model. 

 The Reviewer wrote:

“The calibration process uses gradient descent, however there remains a lack of clarity on the exact implementation used or the various design choices made. The one-line description in the main text is vague and includes the statement, ‘or any of its variants', which is too unspecific for a reproducible work. It would aid the reader to include further details on the implementation. The commercial implementation that is used should also be mentioned directly in the main text (rather than simply referenced). Ideally some implementation details on this could be included in the supplementary text, too. Considering that the commercial company which implemented this is included as an affiliate on the paper, it does not seem unreasonable to expect a bit more details“

Our reply: 

We agree and we have modified the text to add clarifications. We have added in the method section details on the gradient descent:

The calibration process uses the steepest gradient descent to find a combination of rules weights which minimizes the loss function. In our particular implementation we used Limited-memory Broyden–Fletcher–Goldfarb–Shanno algorithm (L-BFGS). 

We have added to the supplementary material that now reads as:

The probability distribution is calibrated to the training set using a cross-entropy loss function. The calibration allows finding the set of weights ω_j which maximizes the likelihood of the observation. The model prediction s for every observation r=〖(r〗_1,r_2,and.,r_n) is computed as the conditional probability of the output variable y: s=P(y=1|r)

The numerical implementation of the calibration routine is based on the Limited-memory Broyden–Fletcher–Goldfarb–Shanno algorithm (L-BFGS). We chose this method for fast convergence and efficient use of computational resources.

We have added the reference to the commercial implementation in the text as well: 

Our model was developed on Stratyfy’s Probabilistic Rule Engine, a commercial machine learning platform26. 

We have added the section on the rule mining algorithm as well as mentioned previously:

To create the forecasting model, the rules were extracted from the available data via rule mining. For this purpose, we used the Association Rule Mining algorithm [32], which is based on the co-occurrences frequency analysis. After extracting the rules, the weights of the rules were determined by the calibration process outlined above.

We have added the reference to paper explaining the underlaying methods for full reproducibility:

32. Jochen Hipp, Ulrich Güntzer, and Gholamreza Nakhaeizadeh. 2000. Algorithms for association rule mining — a general survey and comparison. SIGKDD Explor. Newsl. 2, 1 (June, 2000), 58–64. DOI:https://doi.org/10.1145/360402.360421

 The Reviewer wrote:

“Supplementary material. There is a disconnect between some of the text and the provided tables and figures. Generally, all tables need to be clearly referenced (and include some caption). For example, p12 states, "the learning curve presented in the table,"... yet does not specify which table.“

Our reply: 

We reviewed all the tables and figures and made sure they connect with the text. All tables and figures have their captions submitted as per the guidelines of PLOS one and are shown on the first page of the figures and tables section.

 The Reviewer wrote:

Further comments:

- The cross-entropy function does not render well on the PDF that this reviewer received - several variables were replaced by black rectangles.

Our reply: 

The figures are also sent in high resolution and we believe this will be changed when the final paper is put together. The black rectangles have been removed.

- The references all link to a paperpile.com repository which is not accessible (to this reviewer at least)

Our reply: We have removed the hyperlinks.

- The ROC plots need labels on the axes (e.g. Fig2)

Our reply: 

We have added the labels on the axis

- All tables should include descriptive captions

Our reply: 

We have added the captions in the pdf of submission to address this problem.

The list is the following:

 Figure 1: Data Flow a) the labeling model, b) the forecast model. Each model takes as input three days of data (d, d-1, d-2).

 Figure 2: ROC (False Positive Rate vs True Positive Rate) and Precision/recall curves for the forecasting model with (A, C) and without cognitive assessment (B, D). 

 Figure 3: Model Performance (Learning and Stability). The shade colors represent the variance of results created by the K-Fold(4) validation.

 Table 1: Labeling Model Rules 

 Table 2: Study Populations Demographics 

 Table 3: Algorithm-Derived Rules List of 45 rules extracted by the algorithm and used in the model with their relative weights. Rules are aggregated to forecast suspicion of a viral disease in a participant.

 Table 4: Model Performance With and Without PVT.

- Figure 3 - please specify in the caption what the variance represents

Our reply: 

The variance represents the variance from the 4 runs of validation as per the K-fold validation process. We will add this in the caption 

It now reads as: 

Figure 3: Model Performance (Learning and Stability). The shade colors represent the variance of results created by the K-Fold (4) validation.

Responses to the comments of the second Reviewer:

 The Reviewer wrote:

“The authors have improved the manuscript significantly. The clarity of the manuscript has been significantly enhanced.”

Our reply: We thank the reviewer for its positive feedback.

 The Reviewer wrote:

“I would like the authors to add one final detail in the manuscript.

1. What is the distribution of the labels in data used to train the model. Since, a lot of data is missing due non-compliance I think it is a relevant detail that needs to be added. For eg: If the dataset is skewed towards one class the numbers indicating the performance can be sometimes misleading. As a constant model that predicts a single class all the time can also have high scores on performance metrics. Addition of this data will help increase the confidence on this developed model.”

Our reply: We agree with the reviewer. The distribution of the labels used are too big for 

the main paper but are part of the supplementary materials. We have added text in the discussion about the label distribution of the two classes (positives vs negative) for clarity.

---

## [Decision Letter · Decision Letter 2]

16 Sep 2021

Prediction of viral symptoms using wearable technology and artificial intelligence: A pilot study in healthcare workers

PONE-D-20-36216R2

Dear Dr. DHAESE,

We’re pleased to inform you that your manuscript has been judged scientifically suitable for publication and will be formally accepted for publication once it meets all outstanding technical requirements.

Kind regards,

Nizam Uddin Ahamed, PhD

Academic Editor

PLOS ONE

Additional Editor Comments (optional):

Reviewers' comments:

Reviewer's Responses to Questions

**Comments to the Author**

1. If the authors have adequately addressed your comments raised in a previous round of review and you feel that this manuscript is now acceptable for publication, you may indicate that here to bypass the “Comments to the Author” section, enter your conflict of interest statement in the “Confidential to Editor” section, and submit your "Accept" recommendation.

Reviewer #1: All comments have been addressed

Reviewer #2: All comments have been addressed

2. Is the manuscript technically sound, and do the data support the conclusions?

Reviewer #1: Yes

Reviewer #2: Yes

3. Has the statistical analysis been performed appropriately and rigorously? 

Reviewer #1: Yes

Reviewer #2: Yes

4. Have the authors made all data underlying the findings in their manuscript fully available?

Reviewer #1: No

Reviewer #2: Yes

5. Is the manuscript presented in an intelligible fashion and written in standard English?

Reviewer #1: Yes

Reviewer #2: Yes

6. Review Comments to the Author

Reviewer #1: All reviewer comments have been adequately addressed. I am happy to accept the manuscript in its current form.

Reviewer #2: (No Response)

7. PLOS authors have the option to publish the peer review history of their article (what does this mean?). If published, this will include your full peer review and any attached files.

Reviewer #1: No

Reviewer #2: No

---

## [Editor Report · Acceptance letter]

28 Sep 2021

PONE-D-20-36216R2 

Prediction of viral symptoms using wearable technology and artificial intelligence: A pilot study in healthcare workers 

Dear Dr. D’Haese:

I'm pleased to inform you that your manuscript has been deemed suitable for publication in PLOS ONE. Congratulations! Your manuscript is now with our production department. 

Kind regards, 

on behalf of

Dr. Nizam Uddin Ahamed 

Academic Editor

PLOS ONE